# The Cutaneous Wound Innate Immunological Microenvironment

**DOI:** 10.3390/ijms21228748

**Published:** 2020-11-19

**Authors:** Stephen Kirchner, Vivian Lei, Amanda S. MacLeod

**Affiliations:** 1Department of Dermatology, Duke University, Durham, NC 27707, USA; Stephen.kirchner@duke.edu (S.K.); vivian.lei@duke.edu (V.L.); 2Department of Molecular Genetics and Microbiology, Duke University, Durham, NC 27707, USA; 3School of Medicine, Duke University, Durham, NC 27707, USA; 4Department of Immunology, Duke University, Durham, NC 27707, USA; 5Pinnell Center for Investigative Dermatology, Duke University, Durham, NC 27707, USA

**Keywords:** wounding, skin, innate immunity

## Abstract

The skin represents the first line of defense and innate immune protection against pathogens. Skin normally provides a physical barrier to prevent infection by pathogens; however, wounds, microinjuries, and minor barrier impediments can present open avenues for invasion through the skin. Accordingly, wound repair and protection from invading pathogens are essential processes in successful skin barrier regeneration. To repair and protect wounds, skin promotes the development of a specific and complex immunological microenvironment within and surrounding the disrupted tissue. This immune microenvironment includes both innate and adaptive processes, including immune cell recruitment to the wound and secretion of extracellular factors that can act directly to promote wound closure and wound antimicrobial defense. Recent work has shown that this immune microenvironment also varies according to the specific context of the wound: the microbiome, neuroimmune signaling, environmental effects, and age play roles in altering the innate immune response to wounding. This review will focus on the role of these factors in shaping the cutaneous microenvironment and how this ultimately impacts the immune response to wounding.

## 1. Wounding Induces an Immunological Disruption to the Skin Barrier

The skin is the largest barrier organ by surface area [1] and acts as a vital interface for the body with all manners of the outside world, including microbes and environmental factors. Normal, healthy skin comprises a multilayered barrier that includes the commensal microbiome as well as inherent chemical, physical and immune protectants [1,2]. Each of these components contribute to skin function and integrity, providing avenues for physiological moisture and temperature regulation, ultraviolet (UV) radiation protection, vitamin D production, and perhaps most importantly, keeping pathogenic microbes outside of the body. Despite these barrier components being extremely well suited to their jobs, their homeostatic functions work best when the skin is intact. Wounds, by definition, are disruptors of the skin barrier and not only abruptly change homeostatic skin functions but also provide potential avenues for infection. As such, the skin has developed a robust immunological response to wounding to prevent pathogen invasion into the wound and to resolve and heal the wound. Both innate and adaptive immune responses play roles in this dual process of protection and resolution of wounds. Given the incredibly high burden of skin wounds in the medical system, the skin wound remains an active area of immunological discovery. Recent findings have highlighted the pivotal role of the cutaneous wound innate immunological microenvironment in the physiological response to skin barrier disruption as well as the detrimental role immune cells and their products can play in the setting of chronic wounds and tissue destruction. In this review, we discuss the importance of skin wounds as a clinical issue and highlight work in the innate immune microenvironment of wounds, both host- and environment-derived, that has moved the field towards a more complete understanding of the skin as an immune organ.

## 2. Cutaneous Wounds as Entry Site for Infectious Agents

Cutaneous wounds are a potential avenue for infection, a fact that has been well known throughout history [3]. In the medieval era, both traumatic and iatrogenic wounds had infection rates of close to 80% [3], and even by 1800, 40% of amputations resulted in death, most commonly from sepsis [3,4]. More recently, postoperative wound infection rates in 1988 were found in a World Health Organization study to be as high as 34.4% in some hospitals [5]. In the United States (US), infections still complicate anywhere from 2–5% of surgical sites and are the most common type of hospital acquired infection [6]. Furthermore, this 2–5% may underrepresent actual infection rates by a great deal, as 50% of infections are only apparent after the patient leaves the hospital [7]. Perhaps more alarmingly, patients who develop a postoperative surgical site infection have a mortality risk of 3% [8]. On top of this burden of infectious mortality, surgical site infections pose a huge cost burden to the healthcare system, with an annual expense of USD 3.3 billion in United States hospitals [9].

Furthermore, chronic wounds, defined as wounds that do not go on to heal within 3 months, are unfortunately known as important sources of infection and are incredibly widespread as well: chronic wounds plague an estimated 2% of the US population [10] and are often characterized by dysbiosis, or active microbial infections and biofilm formation, which can negatively impact wound repair. Care for these chronic wounds may cost the US healthcare system USD 28 billion a year [10], and as such, provide an even larger burden on healthcare than their acute counterparts. Clearly, understanding wound infections is important, especially in the context of societal changes that impact wound healing, of which aging [11] and diabetes [12] are of particular relevance.

Wound infections are often considered to be predominately bacterial; many studies have been performed to elucidate the exact microbial colonization of wounds [13]. However, in recent years, a better understanding of the cutaneous wound microbiome has shown that viruses, including bacteria-resident phages, also exploit wounds to promote infection [14]; a specific type of micro-wound, the arthropod bite, is highly permissive to viral transmission. In the US alone, arthropod bites and stings are the cause of thousands of emergency room visits, and between 2010 and 2014, cost roughly USD 7 million to treat in the US [15]. Arthropod bites present opportunities for the introduction of bacterial and viral pathogens as well as parasites to the bite site [16]. Arthropod-derived wound infections are diverse in the US, with both endemic and non-native etiologies [17]. West Nile virus is the leading cause of viral encephalitis in the country [18]. Dengue virus, the most prevalent mosquito-borne viral infection in humans, has increased in incidence almost 30-fold in the last 50 years [19], and Zika virus has similarly seen a sharp rise in case numbers [20]. Despite the skin being the primary site of infection prior to systemic illness in mosquito-borne viral infections, the appreciation and in-depth study of the local skin environment and the immune responses during early infection are lacking.

## 3. Innate Immune Response to Acute Cutaneous Wounds: A Brief Overview

The cutaneous wound is a dynamic immune microenvironment replete with host and invader immunological warfare to close off the wound and defend it from infection. Much of this immune response, from the host perspective, has been well characterized through the years. Wound healing is divided into four phases: 1: hemostasis, 2: inflammation, 3: proliferation, and 4: remodeling [21], which overlap to varying degrees. Excellent reviews of this wound healing process have been previously published [22,23], and as such, we will give a brief overview of the innate immune responses to cutaneous wounds here, in order to better discuss the immune microenvironment in greater detail.

A preliminary inflammatory response to wounding is the formation of a fibrin clot [24], which acts to staunch the flow of blood through the wound and achieve hemostasis. In fact, this hemostatic process is in and of itself an immune phenomenon. Platelets are the primary cellular protagonist of this process and act to promote fibrin cross-linking via the well-studied coagulation cascade leading to thrombin activation [25]. However, this is not how all platelets perform in the wound; platelets are actually active immune effectors [26] and secrete various chemokines, such as C-X-C motif chemokine ligands (CXCL) 1, 4, 5, 7, and 8, as well as antimicrobial defense proteins including beta-defensins 1 and 2 from the clot site [26]. From this perspective, platelets function as an early, first line effector of the immune microenvironment.

Hemostasis occurs in seconds to minutes post-wounding, but in minutes to hours, the wound microenvironment initiates recruitment of professional immune cells upon recognition of danger signals. Upon wounding, skin resident cells, including keratinocytes, dendritic cells, and macrophages, recognize two types of danger signals: damage-associated molecular patterns (DAMPs) and pathogen-associated molecular patterns (PAMPs). DAMPs are self-molecules that arise in the setting of host cell damage, whereas PAMPs are non-self-molecules, such as pathogen specific proteins or nucleic acids, that signal the presence of foreign invaders [27]. PAMPs and DAMPs importantly act, often through toll-like receptors (TLRs), to induce cytokine and chemokine production within the wound. In fact, PAMP signaling plays a key role in wound healing, as TLR4 agonism has been recently shown to shape stem cell tissue repair responses [28]. TLR4 agonism by *Escherichia coli* was also shown to induce antimicrobial defenses in the skin, specifically that of the antimicrobial protein S100A15 [29]; Many other PAMPs and DAMPs also induce antimicrobial host defense molecules [22].

Fibrin clots and platelet activation, along with DAMP/PAMP signaling and the subsequent secreted chemokines [30], lead to further inflammation and recruitment of other immune cells. Neutrophils are the second arrival to the wound bed. Traditionally studied as innate immune cells that act to engulf and kill pathogens invading through the wound [31], neutrophils help to clear pathogens and cellular debris in the wound via engulfment and degranulation, upon which they release a number of destructive enzymes that can act to damage both host and invader [32]. More recently, neutrophils have also been noted to form extracellular traps [33], which act as net-like structures to trap, neutralize and kill pathogens that enter the wound bed. Neutrophil extracellular traps include histones as well as antimicrobial peptides such as defensins and cathelicidins, although more specific roles in the function of traps are yet to be defined [33]. Interestingly, some studies in a human keratinocyte cell line have also shown that neutrophil extracellular traps could induce wound closure in an in vitro scratch assay [34]. However, traps are associated with slower wound healing in diabetes in mouse models [35]. These disparate results elucidate the need for further investigation to better understand how these extracellular factors specifically contribute to the wound microenvironment.

The next immune protagonist in the wound is the monocyte, which traffics to the wound and subsequently dies after activation or differentiates into a macrophage. The role of macrophages in the wound has been extensively reviewed [36]. Macrophages act first in the wound as M1 type pro-inflammatory cells. A subsequent shift in the macrophage population towards an M2 anti-inflammatory phenotype promotes wound healing and resolution. In this way, macrophages function as dual-purposed cells in the immune microenvironment. A wide variety of factors and cytokines are secreted to bridge their two roles: type M1 macrophages are phagocytic and can secrete tumor necrosis factor-α (TNF α), interleukin (IL)-6, IL-12, and IL-1β into the wound environment as pro-inflammatory cytokines, whereas M2 wound-healing macrophages can secrete factors such as transforming growth factor-β (TGF-β) and IL-10, among others [36]. The importance of these dual roles in wound defense and healing processes are exemplified particularly in the diabetic wound model, where the balance and regulation of M1 and M2 phenotypes is altered. Multiple studies of diabetic wounds have suggested that a skew towards a longer duration of the M1 phenotype leads to chronic inflammation and lack of wound healing and resolution in diabetic skin [37,38,39].

The immune landscape in acute cutaneous wounds is complex and a further layer of complexity is added to the innate immune response when we are asked to consider each wound’s specific microenvironment, which has been a recently active area of research. Aspects of this microenvironment, including neuroimmune, microbial, and environmental factors, all impact the innate immune cell milieu and contribution to wound responses. The rest of this review will be dedicated to these specific aspects that can alter the innate immune microenvironment of wounds.

## 4. Sensing a Wound: Immune Microenvironment Is Dependent on Neuroimmune Signaling

One aspect of any wound that impacts how the immune microenvironment is formed is the neuroimmune axis. Skin is innervated by a distinct neuronal network [40] that influences skin physiological responses, as well as the pathophysiological response with respect to wounding. Upon skin wounding, neurons sense the disruption and can release neurotransmitters into the wound microenvironment. Among these are neurotransmitters such as substance P, calcitonin gene related peptide (CGRP) and galanin (GAL) [40]. A full discussion of the functions of these neurotransmitters is outside the scope of this review; however, many neurotransmitters have immune-stimulating effects to influence cytokine production and immune cell recruitment [40]. It should also be noted that neuronal sensing and neuropeptides play key roles in vascular supply [41], which is critical to proper immune cell trafficking and healing of the wound. Furthermore, neural sensing has been shown to be critical in wound healing, with denervated rat skin displaying reduced rates of wound contraction [42], and chemical denervation of skin reducing the inflammatory cell infiltration upon wounding [43]. More recently, specific crosstalk between neurons and the skin immune cell response has been elucidated. Nociceptive (noxious stimuli sensing) receptors in the skin were found by one group to activate CD301b^+^ dendritic cells via the neurotransmitter CGRP to produce IL-23 in the context of *Candida albicans* pathogen challenge [44]. This group further went on to characterize that these nerve fibers, defined by the cation channel transient receptor potential cation channel subfamily V member 1 (TRPV1), can directly activate skin host defenses and lead to an increase in neutrophil and lymphocytic recruitment to areas of neuronal activity [45]. This was characterized as a type 17 immune response, and type 17 immunity has previously been linked to antimicrobial peptide production in wounds [45,46]. While these studies together have unveiled that neuronal activation plays a key role in skin immune cell recruitment and activation, the specifics of how wounding, and the nociception involved, leads to specific macrophage or dendritic cell activation and subsequent host defense and wound closure are still incompletely characterized. As nociceptive fibers activate CD301b^+^ dendritic cells [44], which have documented roles in antiviral defense and wound healing via production of the cytokine IL-27 and additional factors [47,48,49], it is possible that wounding directly activates a pain-mediated dendritic cell response to protect and close barrier disruptions.

## 5. Host–Microbe Interactions in the Wound as a Component of the Immune Microenvironment

The skin is an ecosystem, home to both host cells of various types, but also to skin-resident bacteria, fungi and viruses [50]. Much of this microbiome is commensal but parts of it can also be highly pathogenic. Intriguingly, numerous studies have shown that wounding can dramatically alter the cutaneous microbiome [51]; the disruption of skin can lead to bacteria taking advantage of a new environment, a phenomenon termed quorum sensing. This often leads to a reduction in bacterial diversity in chronic wounds [52], as well as in acute traumatic fracture wounds [53]. Furthermore, in acute traumatic fracture wounds, wounded tissue has an initially distinct microbiome from nonwounded tissue, but eventually the two converge over the course of healing [53], suggesting a dynamic role of the microbiome in the wound resolution process. Significant work has been performed to elucidate how skin flora may either potentiate or impair wound immune responses (Figure 1). One such study compared germ-free (gnotobiotic) Swiss mice that had no commensal microbiota to conventionally raised Swiss mice [54] and found that germ-free mice not only heal faster than conventional mice, but also recruit fewer neutrophils as well as more mast cells and macrophages to the wound tissue [54]. These findings could be reversed when germ-free mice were conventionalized with the microbiota of normally raised mice [54]. This result is understandable, as a skin wound needs to both protect and heal, particularly with respect to macrophage contributions to this local immune response. While this study displayed an overall effect of the microbiome on immune cell recruitment and the subsequent wound healing response, more specific studies on skin commensals and pathogens have also demonstrated their importance to the immune microenvironment of wounds. For instance, the classic commensal bacteria of the skin, *Staphylococcus epidermidis*, has recently been shown to engender a non-classical major histocompatibility complex 1 (MHC1) T cell response [55]. These T cells exhibit an effector signature but also an immunoregulatory and tissue-repair signature that ultimately leads to wound healing in mice [55]. Furthermore, the *S. epidermidis* component lipoteichoic acid modulates inflammation in response to wounding [56] via TLR2-induced Tumor necrosis factor receptor associated factor (TRAF) 1 inhibition of TLR3. *Staphylococcus epidermidis* also acts directly to promote host immune defenses via the production of phenol soluble modulins that exert antimicrobial effects against skin pathogens [57]. This, combined with the germ-free data [54], suggests that distinct bacteria, even within the normal commensal population, exert differing effects on the wound immune response and wound healing outcome.

The composition of the skin bacteriome might explain some of the differences between healing and non-healing wounds. Bacterial DNA profiles were found to differ vastly between wounds that go on to heal and those that do not [62], as does expression of the pattern recognition receptor nucleotide-binding oligomerization domain-containing protein 2 (NOD2). NOD2 deficiency was found to produce an altered microbiome, with an increased level of pathogenic *Pseudomonas* species and a trend towards decreased commensal incidence (*S. epidermidis*) [61] and associated wound repair delays [63]. Chronic, non-healing wounds have been found in several studies to have differing microbiomes to healing wounds [64,65]; however, more work needs to be undertaken to understand exactly why this is the case. One possibility is pathogenic immune evasion and active suppression. *Staphylococcus aureus*, a common pathogen cultured from non-healing wounds, can promote lysis of macrophages, neutrophils, and monocytes via beta-barrel forming toxins [58]. In the case of *Pseudomonas*, biofilm formation [66] may play a key role in preventing ultimate wound closure and proper immune cell infiltration to the wound site. Nevertheless, greater understanding of how the microbiome influences the immune responses at the site of wounds is needed, which may in turn help us clinically promote a healthy microbiome to improve wound responses.

Most publications linking the microbiome to wound immune responses have understandably focused on bacteria, given their high infectious burden. However, this ignores both skin resident fungi and viruses, which also significantly contribute to skin homeostasis. Fungal communities have been shown to be predictive of healing time, with the phylum *Ascomycota* being proportionally higher in slower healing cutaneous wounds compared to faster healing wounds [67]. Mechanistically, one common phylotype of this phylum, *Candida albicans*, has the capacity to impair macrophage function and actually kill these immune cells [59]. If the innate immune system is unable to clear fungal pathogens in chronic wounds, these wounds may become a hotspot of continued inflammation. As such, consideration to the fungal components of the immune microenvironment must be made when attempting to understand skin wounds.

A small number of studies have examined and shown the skin virome [68,69] to be a significant component of the skin ecosystem. Studies on viral impact to the wound microenvironment are also not numerous, although there is some evidence that skin-tropic papillomaviruses [60] hijack wound healing [60] and preferentially infect wounded keratinocytes [70]. However, how skin-tropic viruses contribute to immune responses in wounded tissue requires further study. Interestingly, a recently discovered pathway involving production of the cytokine IL-27 [71] from CD301b+ immune cells in wounded tissue activates antiviral defenses and promotes wound healing [47]. The antiviral defenses induced by IL-27 were found to be signal transducer and activator of transcription 1 (STAT1)-dependent but STAT2-independent [72] in a fashion that mediated Zika virus immunity in the skin. These findings would place IL-27 very much at the immune intersection of skin wound protection and healing; however, further studies to understand this pathway are needed, specifically what lies upstream of IL-27 in the wound immune response. Nevertheless, these findings suggest that viral components of both host defense and pathogens could play a role in wound immune microenvironments, and, together with the known findings of the skin microbiome and fungal components, show that skin immune responses to wounding are informed greatly by the microorganisms present within the wound environment.

## 6. Environmental Effects: The Outside World Alters the Cutaneous Wound Environment

So far, discussion of the wound immune microenvironment has been limited to local host or microbial contributions. However, increasingly, we are understanding that immune responses to wounds are also shaped by the external environment (Table 1). Naturally, outside influences affect our skin greatly, as skin is the main interface barrier with the outside world. As such, important, but often overlooked, skin variables that can greatly impact the cutaneous wound environment include moisture levels, UV radiation, wound timing, and location of the wound on the body (Figure 2).

Our skin naturally loses roughly half a liter [73] of water a day via transepidermal water loss. However, this amount can change dramatically as a function of the outside world [74]. It has been known for years that skin moisture impacts wound re-epithelialization, but recent work has actually determined the exact levels of moisture that are most beneficial for wound closure [75]. Intriguingly, skin moisture has been shown to also impact immune responses of the skin; the occlusive emollient petrolatum can induce antimicrobial defenses in the skin including human beta defensin 2, lipocalin 2, as well as chemokine ligands CXCL1 and CXCL2 [76]. This effect was stronger with petrolatum than with standard occlusion [76], which suggests that skin moisture status regulates innate defenses that have not only antimicrobial but also wound healing effects [77,78]. Given that moisture also promotes immune cell infiltration into a wound [79], it is clear that skin hydration plays a pivotal role in the immune microenvironment of wounds.

**Table 1 ijms-21-08748-t001:** Key factors in the innate immune microenvironment that impact wounds.

	Microenvironment Component	Outcome(s)	Reference(s)
**Internal**	Neural Sensation	Denervated skin heals at slower rates;	[42]
		TRPV1 nerve fibers activate host immune defenses	[44,45]
**Internal**	Wound Location	Immune cell numbers vary with body site	[80]
**Internal**	Age	Elderly Skin heals slower than younger skin;	[81]
		Inflammation/repair spectrum is impaired in aged skin	[82]
**External**	Cutaneous Bacteria	Microbiome deletion potentiates wound closure;	[54]
		Commensal microbes can promote antimicrobial defense;	[57]
		Microbiome is altered in chronic, non-healing wounds	[64,65]
**External**	Cutaneous Fungus	Cutaneous fungal communities are predictive of wound healing time	[67]
**External**	Cutaneous Virus	IL-27 promotes antiviral defense and healing in cutaneous wounds	[47]
**External**	Moisture	Emollients can promote antibacterial defenses;	[76]
		Skin moisture levels directly impact wound healing rate	[75]
**External**	UV Radiation	UVB radiation activates Type I interferon responses;	[83]
		UVB radiation can directly stimulate wound healing	[84]
**External**	Time of Wound	Fibroblast migration and wound healing varies with time of wound	[85]

**Figure 2 ijms-21-08748-f002:**
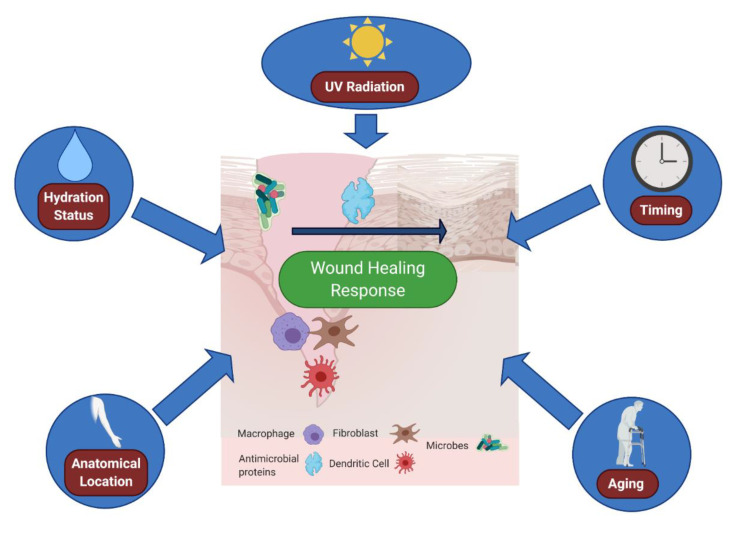
Variables affecting the cutaneous immune microenvironment of wounds. Environmental impacts, such as skin moisture, ultraviolet (UV) exposure, timing of wounding and location of the wound, can all modulate immune responses to barrier disruption. Skin moisture levels can alter immune expression of antimicrobial proteins [76] as well as immune cell infiltration [79]. UV exposure can promote wound closure [84], as well as interferon signature [83]. Time of day can alter fibroblast activity in the wound [85], as well as immune cell trafficking [86]. The location of the wound on the body matters as well, as various immune cells differ in proportion throughout the body dependent on the location [80]. Finally, aging plays a major role in the microenvironment of wounds; aged skin is deficient in neutrophil and macrophage recruitment [87] and is also slower to re-epithelialize [88]. Figure created using Biorender.

Yet another environmental factor that can impact cutaneous wound immune components is that of UV radiation. One study in 2012 was able to demonstrate that UV-C band radiation could increase wound healing rates of murine skin wounds infected with *S. aureus* [89], likely due to direct reduction in bacterial burden. UV-C radiation is blocked by ozone molecules, so is likely not a major component of human environmental exposure [84]. However, UV radiation also has well documented effects on human immune responses and appears to be a double-edged sword in the wound immune response. UV radiation of B wavelength only needs one dose to actively induce murine skin to produce type I interferon, as well a number of distinct antiviral genes including interferon regulatory transcription factor 7 (Irf7), interferon induced protein with tetratricopeptide repeats 1 (Ifit1), interferon stimulated gene 15 (Isg15), and myxovirus resistance 1 (Mx1) [83]. Induction of these genes can possibly confer a protective, antiviral state to the immune microenvironment. Moreover, type I interferon receptor deficiency slows wound re-epithelialization in response to a tape strip injury, likely due to a decrease in downstream immunoregulatory cytokines, including IL-6, IL-17, and IL-22 [90]. UV radiation’s ability to induce interferons and the antimicrobial and wound-healing roles of interferons could explain some of UV radiation’s ability to enhance wound recovery [84]. However, UV radiation can also be immunosuppressive at different doses, as additional studies have also shown impaired wound closure in UV-irradiated skin [91]. These works demonstrate an incomplete understanding of the roles of UV on the cutaneous immune landscape, despite its long-term study in the field. It is possible that UV’s effect on the wound varies according to the time of irradiation within the immunological phases of the wound. For instance, UV radiation promotes neutrophil extracellular trap formation [92], which, as previously discussed, is a specific neutrophil defense that is protective against pathogens but also highly destructive to host tissue [33]. Recent findings of the skin microbiome’s interaction with UV add another layer of complexity: the skin microbiome actually modulates UV-induced immune suppression of the skin [93].

Another component of the environment that is linked to altered wound responses is that of timing. A recent publication [85] has shown that the time of day impacted actin regulation in fibroblasts, which in skin wounds led to altered fibroblast invasion into the wound dependent on the time of day. This was supported by data from human burns, which healed at different rates depending on time inflicted. The researchers tied this directly to the circadian clock [85], which coordinates rhythmic activity throughout the body. Circadian regeneration of tissue in the skin [94] has been studied from the fibroblast and keratinocyte responses in cutaneous wounds but little work to date has been performed to evaluate the immune contributions to time-stamped wounds. Circadian clock components, such as the transcription factor Brain and Muscle ARNT-like 1 (BMAL1), have been shown to play key roles in innate immune responses [95], including in macrophages where they have recently been shown to maintain mitochondrial metabolism under stress [96]. Intriguingly, skin responses to inflammatory stimuli, such as the toll-like receptor 7 agonist imiquimod, are varied throughout the day [97]. However, it remains to be seen if the wound recruitment and ultimate response of the innate immune system are specifically time of day regulated, and if so, in what way. Given that leukocyte numbers and trafficking are time of day controlled [86], it is possible that the immune cell infiltration of the wound is variable with wound timing. Even more specifically, cytokine secretion [98] is circadian gated in other non-skin tissues, and so understanding if this is true in wounds would greatly shape our understanding of the immune microenvironment of this disrupted tissue.

Not only is the timing that a skin wound is inflicted important, but the location can also impact the immune microenvironment. One simple reason for this is skin thickness, which can vary widely throughout the body, even in the human face alone, epidermis ranges from 62.6 micrometers at its thickest to 29.6 micrometers at its thinnest [99]. This variable thickness leads to a distinction in skin wounds, namely whether a wound disrupts solely epidermis, or epidermis and deeper dermal structures. In the first case where only the epidermis is disrupted, also called partial thickness wounding [100], wounds are healed primarily by re-epithelialization alone [100]. This is opposed to full-thickness wounds, with dermal damage, which require granulation tissue formation [100]. Fibroblasts have been known to be the primary protagonist in the creation of granulation tissue [100], but only recently has it been determined that up to two-thirds of fibroblasts in wound tissue are derived from myeloid cells [101]. Intriguingly, cluster of differentiation 1a (CD1a)-positive Langerhan cells, a skin specific immune cell [102], as well as CD86-positive macrophages, were found to vary in quantity according to skin biopsy site [80]. This could partially explain variable healing rates in different anatomical locations. Anatomical location also impacts the skin microbiome [50], with widely differing bacterial compositions on different skin sites dependent on the skin’s sebaceous, moist, or dry environment. While it appears that the bacteria and static innate immune populations of the skin change according to location, it remains unclear if immune cell trafficking follows a similar pattern. Blood supply to the skin varies with location [103], and it is certainly possible that the capacity for immune cell transport to the site of the wound is dependent on the vascular perfusion of the skin site. More work on understanding this may be crucial, particularly for the understanding of chronic wounds and how to best treat them based on the location of the tissue disruption.

## 7. An Aging Microenvironment

One final aspect of immune responses to wounding that cannot be overlooked is that of aging. It has been known for some time that elderly individuals heal wounds more slowly than their younger counterparts [81,104], but the exact mechanisms behind this phenomenon have garnered some recent attention. Aged skin has difficulties in all phases of the wound response [105], but specifically with respect to immune responses. Both intracellular and vascular cellular adhesion molecule-1 (CAM-1) had altered profiles in aged skin wounds [106], leading to a differential temporal immune cell infiltration pattern to younger skin. Another aspect of the skin wound that changes with age is hormonal. Notably, estrogen, which can be deficient in aging, can alter the immune response and potentiate wound repair [107,108]. More recent work in the realm of wound research has shown that elderly wounded skin tissue has diminished neutrophil and macrophage recruitment in mice [87], which could play a role in both wound infections and wound healing for the elderly. Other work in this realm has shown age to play a key role in the expression of a number of wound-related genes in mice, including TGF-Beta, MCP-1, MMP9, and MMP13, resulting in differential rates of wound re-epithelialization between young and old mice [88]. With the rise of sequencing-based technologies, we have also learned much about the expression profiles of wounded epidermal skin across the age spectrum. Using RNA-Seq on wounded murine keratinocytes, a publication in 2016 called attention to the downregulation of a number of immune function genes in aged skin, including *Il6*, *Il10*, *Il7*, and *Defb1* [109]. Intriguingly, this work went on to characterize that dendritic epidermal T-cells [109] were a crucial immune cell actor in wound re-epithelialization, which was disrupted in aged-skin wounds. This would suggest elderly skin does not appropriately activate immune responses to wounding; however, this contrasts with other works that suggest elderly skin is skewed towards inflammation. Using RNA-Seq in intact human skin, one group displayed that increasing age was associated with an increase in gene transcription of inflammatory pathways but not with a corresponding increase in repair processes [82], possibly playing a role in the wound healing disruption seen in elderly populations. More weight to this argument was given by a single cell RNA-Seq publication examining populations of fibroblasts in both young and aged human skin [110]. Fibroblasts, as a key protagonist in the wound closure phase, necessarily crosstalk with immune cells in wounds. However, in aged skin, fibroblast interaction with macrophages and dendritic cells is perturbed [110]. Furthermore, aged fibroblasts express more pro-inflammatory cytokines, including CXCL2, CXCL14, CXCL1, CXCL3, and IL-32 when compared to their younger counterparts [110]. As such, it appears from this collection of work that a slant towards inflammation may disrupt normal wound repair in aged skin. However, skin and soft tissue infections still happen at a higher rate in adults over 65 than in middle-aged adults [111], indicating that this inflammatory skew in elderly skin is still not protective. This creates a relative disconnect between increased inflammatory pathways in skin cells but possibly less professional immune cells in aged wounds.

As our understanding of aging skin and the wound response evolves, so does our targeted therapies. For example, platelet-rich plasma (PRP) therapy presents a promising treatment modality for aged wounds [112]. PRP is noted to promote fibroblast proliferation and tissue remodeling crucial to wound repair. Given that platelet count is thought to decline with age [113], it may be possible that replacing these immune cells with PRP reverts the wound environment to a younger, healthier state. These findings show that immune factors can play a major role in aged skin repair. As such, further research is needed to understand how aged skin’s altered immune microenvironment is created and maintained over the lifespan, and what drivers of this effect could be targets for therapies.

## 8. Conclusions

As this review attempts to address, not only are we constantly learning more about the specifics of the immune response to wounding, we are also developing a more complete understanding of the immune microenvironment in which wounds exist (Table 1). Every wound has a unique context, and with that context, there is a unique microenvironment defined by microbial composition, neuronal sensation, environment, and host age. How our skin immunologically responds to wounds varies with respect to all of these facets, making the skin wound incredibly variable and complex as an immunological research landscape. Wound care, either in an acute or chronic setting, is a major clinical care issue worldwide. Understanding how the immune microenvironment shapes wound responses, both in terms of host defense and wound resolution, is crucial to betterment of basic and clinical science. With a more comprehensive view of the immune microenvironment, we can better tailor our efforts to wound care, keeping wounds from infection and ultimately leading them to heal.

## Figures and Tables

**Figure 1 ijms-21-08748-f001:**
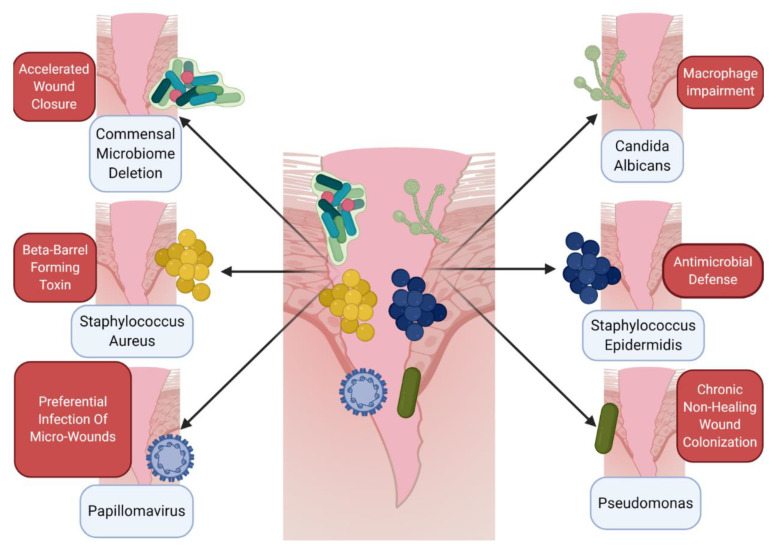
Host–microbe cross talk informs the immune microenvironment of wounds. A number of different microbes, including various bacteria, viruses, and fungi, can directly impact wound healing and alter the immune microenvironment. Upon wounding, whether by trauma, arthropod bite, or chronic barrier disruption, skin-resident bacteria, viruses, and fungi all can impact the microenvironment of the healing tissue. Several general and specific examples of this microenvironment interaction have been elucidated recently. Paradoxically, germ-free mice exhibit accelerated wound closure when compared to conventionally raised mice [54]. Other specific microbes have distinct effects on immune cells, including *Staphylococcus aureus*, which creates beta-barrel forming toxins to impair macrophage function in the wound [58], or *Candida albicans*, which also disrupts normal macrophage action [59]. While not a direct immune cell effect, other pathogens can impact the microenvironment, hijacking or colonizing wounds directly, including papillomavirus [60] and *Pseudomonas* species [61]. Skin commensal microbes such as *Staphylococcus epidermidis* can promote immune defenses in the wound [57] via phenol-soluble modulins. This provides a snapshot of the various roles of microbes in the wound microenvironment but is not exhaustive. Further study is needed to fully understand the skin microbiome–wound microenvironment interaction. Figure created using BioRender.

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
