# Peer review of "The Cutaneous Wound Innate Immunological Microenvironment"

_ijms, 2020, doi:10.3390/ijms21228748_

Round 1

Reviewer 1 Report

The review manuscript by Stephen Kirchner and co-authors is well written, narrative review that has provided a brief overview of immune environment in the skin that keeps and restores the local homeostasis in balance. The topic of the paper is interesting and its content is acceptable. However, it will be necessary for authors to undertake revisions per the comments.

  • The title of the paragtaph 2 („Regeneration and Restoration…”) does not mach the content of the text. As there is no information of wound repair or regeneration proces in the text, the heading should be modified.
  • It is expected that the authors add a table with the home-take massages and their references.
  • The following article demonstrates the expression of CD68, MCP1 and Mac-2 in the young and old mice during cutanous wound healing:

https://www.aging-us.com/article/103064/text

Please consider to use these data in paragraph 7 („An Aging Microenvironment”).

Author Response

Reviewer 1:

The review manuscript by Stephen Kirchner and co-authors is well written, narrative review that has provided a brief overview of immune environment in the skin that keeps and restores the local homeostasis in balance. The topic of the paper is interesting and its content is acceptable. However, it will be necessary for authors to undertake revisions per the comments.

The title of the paragraph 2 („Regeneration and Restoration…”) does not mach the content of the text. As there is no information of wound repair or regeneration proces in the text, the heading should be modified.

We thank the reviewer for this edit; please note in line 64 that the new heading for this paragraph is “Cutaneous Wounds As Entry Site for Infectious Agents”.

It is expected that the authors add a table with the home-take massages and their references.

Thank you for this suggestion; we have added a table with take-home messages and references in lines 433-434. 

The following article demonstrates the expression of CD68, MCP1 and Mac-2 in the young and old mice during cutanous wound healing:https://www.aging-us.com/article/103064/text Please consider to use these data in paragraph 7 („An Aging Microenvironment”).

Thank you for alerting us to this publication; it has been cited and discussed in lines 395-398.

Reviewer 2 Report

Excellent review on a complex and dynamic topic. Hopefully the review will inspire others to explore the immunodiversity of the skin and its impact after reading your paper. Just a few suggestions to strengthen the manuscript.

Section 2. You cover SSI (ie acute wound) prevalence, costs and mortality rates as well as the arthropod bites and viral transmission vector costs. You did not however, address the chronic wound  prevalence, costs and mortality rates and the polymicrobic nature of these wounds. Understandably you could use the entire review on just that type as its infection and immunity processes are so different than the acute surgical wound, but you do the reader a disservice for not providing a brief synopsis about the chronic wound issues that are far more deadly and expensive. 

Section 3. In the last paragraph of this section, it would be good to let the reader know you are specifically discussing the acute wound setting since you eluded to some of the complexities involved with diabetic wounds in the last sentence of the previous paragraph. You may even want to further clarify this by including the word "acute wound" within your title. 

Section 4. We must remember that not only can the neurons elicit inflammation and immune response, but it also impacts the vascular system locally which can tremendously impact the microenvironment, all of which either contributes or is detrimental to the rate of healing.  

Section 5. The hemostasis of the host and the natural microbiome is interrupted when the skin is wounded. The microbiome works with the keratinocytes to secrete cytokine and neurotransmitters as well upon wounding working in conjunction with the commensal induction of immune cells from the bacteria. Lastly, the natural microbiome is diverse and tends to stay in check via both the host immune cells but also through quorum sensing. Once the microbiome is interrupted, opportunistic bacteria can try to "expand their claim" as well as aggressive pathogens trying to get a foothold in the wound and skin. There is a lot going on and your figures illustrate a lot of it quite well, but I think you could expand this out a bit more.

Section 7. There's several excellent studies that are relevant to aging and wound healing to consider including in this review. In young and aged animal wounding studies, healing rate of the aged animals could be sped up to match the younger counterparts when given a topical application of the young animal buffy coat from a blood draw (think pre PRP therapy). There's also a variety of PRP treatment manuscripts and one theory is the concentration of immune cells to the wound in an aged person is more than would normally be present in the wound. Lastly, there's the hormonal impact of the aged skin compared to younger counterparts.

Author Response

Excellent review on a complex and dynamic topic. Hopefully the review will inspire others to explore the immunodiversity of the skin and its impact after reading your paper. Just a few suggestions to strengthen the manuscript.

Section 2. You cover SSI (ie acute wound) prevalence, costs and mortality rates as well as the arthropod bites and viral transmission vector costs. You did not however, address the chronic wound  prevalence, costs and mortality rates and the polymicrobic nature of these wounds. Understandably you could use the entire review on just that type as its infection and immunity processes are so different than the acute surgical wound, but you do the reader a disservice for not providing a brief synopsis about the chronic wound issues that are far more deadly and expensive. 

Thank you for comments on covering acute and chronic wounds; please note lines 58-59 and 76-81 which now mention and discuss chronic wounds. We have now added into the text: “Recent findings have highlighted the pivotal role of the cutaneous wound innate immunological microenvironment in the physiological response to skin barrier disruption as well as the detrimental role immune cells and their products can play in the setting of chronic wounds and tissue destruction”   and  “Furthermore, chronic wounds defined as wounds that do not go on to heal within 3 months, are unfortunately known as important sources of infection and incredibly widespread as well: chronic wounds plague an estimated 2% of the US population [10] and are often characterized by dysbiosis, or active microbial infections and biofilm formation, that can negatively impact wound repair. Care for these chronic wounds may cost the US healthcare system $28 billion a year [10], and as such, provide an even larger burden on healthcare than their acute counterparts. Clearly, understanding wound infections is important, especially in the context of societal changes that impact wound healing, of which aging [11] and diabetes [12] are of particular relevance.” The latter section had been revised also in response to comments by reviewer 3. Thus we believe that we addressed the comments of this reviewer appropriately by providing a brief synopsis of issues for these wounds.

Section 3. In the last paragraph of this section, it would be good to let the reader know you are specifically discussing the acute wound setting since you eluded to some of the complexities involved with diabetic wounds in the last sentence of the previous paragraph. You may even want to further clarify this by including the word "acute wound" within your title. 

We agree with the reviewer on this clarification; please note the changed title in line 100, and the addition of ‘acute’ in line 158 to “Innate Immune Response To Acute Cutaneous Wounds: A Brief Overview.”

Section 4. We must remember that not only can the neurons elicit inflammation and immune response, but it also impacts the vascular system locally which can tremendously impact the microenvironment, all of which either contributes or is detrimental to the rate of healing.  

Thank you-This suggestion by the reviewer is well taken, and has been acknowledged in lines 173-175, with a reference added to discuss neurological control of vascular supply. “It should also be noted that neuronal sensing and neuropeptides play key roles in vascular supply [41], which is critical to proper immune cell trafficking and healing of the wound.”

Section 5. The hemostasis of the host and the natural microbiome is interrupted when the skin is wounded. The microbiome works with the keratinocytes to secrete cytokine and neurotransmitters as well upon wounding working in conjunction with the commensal induction of immune cells from the bacteria. Lastly, the natural microbiome is diverse and tends to stay in check via both the host immune cells but also through quorum sensing. Once the microbiome is interrupted, opportunistic bacteria can try to "expand their claim" as well as aggressive pathogens trying to get a foothold in the wound and skin. There is a lot going on and your figures illustrate a lot of it quite well, but I think you could expand this out a bit more.

Thank you for the suggestion and this excellent point. We agree with the reviewer that these aspects of the natural microbiome disruption in wounding could be expanded upon. Please note the changes made in lines 198-205, detailing how wounding can directly alter the natural microbiome, before we return to more specifics of microbial community impacts on the wound.

Section 7. There's several excellent studies that are relevant to aging and wound healing to consider including in this review. In young and aged animal wounding studies, healing rate of the aged animals could be sped up to match the younger counterparts when given a topical application of the young animal buffy coat from a blood draw (think pre PRP therapy). There's also a variety of PRP treatment manuscripts and one theory is the concentration of immune cells to the wound in an aged person is more than would normally be present in the wound. Lastly, there's the hormonal impact of the aged skin compared to younger counterparts.

We appreciate the reviewers’ comments on this aging section, which we feel has been strengthened by their suggestions. To the reviewer’s comment on hormonal impact, please note lines 391-393, which provide a brief discussion of estrogen effects on aged wound healing. Furthermore, the discussion of PRP and platelet treatments to improve aged skin healing has been incorporated into lines 420-427, which we believe fits well with the discussion of current and future therapies for wound healing.

Reviewer 3 Report

Please make more concise or remove Section 2 (regeneration and restoration following ...) to focus on the innate immunological microenvironment of the wound.

Please supplement more detailed information regarding section 6 (environmental effects) in figure 2.

There are some typos in the manuscript, for example, (page 4 line 155) galanoin -> galanin, (page 9 line 374) RN-Seq -> RNA-Seq.

Author Response

Please make more concise or remove Section 2 (regeneration and restoration following ...) to focus on the innate immunological microenvironment of the wound.

We thank the reviewer for this suggestion; we have attempted to make parts of this section more concise through several deletions and simplifications. However, we would like to note that reviewer 2 suggested the addition of information regarding chronic wounds, and as such, this has been added.

Please supplement more detailed information regarding section 6 (environmental effects) in figure 2.

We appreciate this commentary on Figure 2; we have attempted to provide more detailed information in the Figure 2 legend, in a more similar fashion to the Figure 1 legend.

There are some typos in the manuscript, for example, (page 4 line 155) galanoin -> galanin, (page 9 line 374) RN-Seq -> RNA-Seq.

We appreciate the proofreading eyes of the reviewer; these typos have been corrected.